# Dramatic Suppression of Lipogenesis and No Increase in Beta-Oxidation Gene Expression Are among the Key Effects of Bergamot Flavonoids in Fatty Liver Disease

**DOI:** 10.3390/antiox13070766

**Published:** 2024-06-25

**Authors:** Maddalena Parafati, Daniele La Russa, Antonella Lascala, Francesco Crupi, Concetta Riillo, Bartosz Fotschki, Vincenzo Mollace, Elzbieta Janda

**Affiliations:** 1Department of Health Sciences, Magna Graecia University, Campus Germaneto, 88100 Catanzaro, Italy; mparafati@unicz.it (M.P.); francesco.crupi@unicz.it (F.C.); concettariillo@unicz.it (C.R.); mollace@unicz.it (V.M.); 2Department of Biology, Ecology and Earth Sciences, University of Calabria, 87036 Rende, Italy; daniele.larussa@unical.it; 3Department of Biological Function of Food, Institute of Animal Reproduction and Food Research, Polish Academy of Sciences, 10-748 Olsztyn, Poland; b.fotschki@pan.olsztyn.pl

**Keywords:** hepatic steatosis, lipogenesis, *Citrus bergamia*, gene profiling, flavonoids, flavanones, lipid synthesis

## Abstract

Bergamot flavonoids have been shown to prevent metabolic syndrome, non-alcoholic fatty liver disease (NAFLD) and stimulate autophagy in animal models and patients. To investigate further the mechanism of polyphenol-dependent effects, we performed a RT2-PCR array analysis on 168 metabolism, transport and autophagy-related genes expressed in rat livers exposed for 14 weeks to different diets: standard, cafeteria (CAF) and CAF diet supplemented with 50 mg/kg of bergamot polyphenol fraction (BPF). CAF diet caused a strong upregulation of gluconeogenesis pathway (*Gck*, *Pck2*) and a moderate (>1.7 fold) induction of genes regulating lipogenesis (*Srebf1*, *Pparg*, *Xbp1*), lipid and cholesterol transport or lipolysis (*Fabp3*, *Apoa1*, *Lpl*) and inflammation (*Il6*, *Il10*, *Tnf*). However, only one β-oxidation gene (*Cpt1a*) and a few autophagy genes were differentially expressed in CAF rats compared to controls. While most of these transcripts were significantly modulated by BPF, we observed a particularly potent effect on lipogenesis genes, like *Acly*, *Acaca* and *Fasn*, which were suppressed far below the mRNA levels of control livers as confirmed by alternative primers-based RT2-PCR analysis and western blotting. These effects were accompanied by downregulation of pro-inflammatory cytokines (*Il6*, *Tnfa*, and *Il10*) and diabetes-related genes. Few autophagy (*Map1Lc3a*, *Dapk*) and no β-oxidation gene expression changes were observed compared to CAF group. In conclusion, chronic BPF supplementation efficiently prevents NAFLD by modulating hepatic energy metabolism and inflammation gene expression programs, with no effect on β-oxidation, but profound suppression of de novo lipogenesis.

## 1. Introduction

Non-alcoholic fatty liver disease (NAFLD) and its more advanced form (NASH), are the most common liver disorders in industrialized countries, caused by fat and sugar-rich diet, sedentary lifestyle and genetic predisposition [1,2,3]. To date, there are no specific drugs approved for NAFLD [4,5], but antioxidant polyphenols, especially in the form of plant-derived extracts, are emerging as an important therapeutic option for the management of NAFLD and NASH [6] in addition to dietary measures and physical activity. In this scenario, bergamot polyphenol fraction (BPF) appears as a particularly promising food supplement. BPF^®^ obtained from the juice and peels of bergamot (*Citrus bergamia* Risso et Poiteau) fruits exceptionally rich in flavonoids, is characterized by a unique profile of flavonoids such as naringin, brutieridin and melitidin and many other flavonoid and non-flavonoid compounds with lipid-lowering, anti-inflammatory, proautophagic and detoxifying activity [7,8,9,10]. The chemical composition and beneficial effects of BPF antioxidants have been documented in numerous analytical [7,8,11], preclinical [9,12,13,14,15] and clinical studies [10,16,17,18,19,20]. In particular, vast evidence proves the efficacy of BPF against diet-induced NAFLD and NASH in rodent models [9,13,15] and in clinical studies [18], but little is known about molecular mechanisms underlying these effects.

Hepatic lipid accumulation is a product of an imbalance between fatty acids (FA) synthesis and FA oxidation, but oxidative stress and inflammation also play a key role in steatosis pathophysiology [21]. FA synthesis is the first step of de novo lipogenesis (DNL). It starts with the conversion of citrate or acetate to acetyl-CoA by the action of citrate lyase (ACLY) or hepatic ACSS2, respectively [22]. Acetyl-CoA is then converted to malonyl-CoA by acetyl-CoA carboxylase (ACACA). FA synthase (FASN) sequentially utilizes malonyl-CoA to extend the growing fatty acyl chains. Most lipogenic enzymes are upregulated in high-fat diet [23,24,25], fructose or sucrose [22,26,27] or Western diet-induced [28] animal models of NAFLD and NASH, but their differential expression has never been investigated in CAF diet compared to normal diet livers.

DNL is also tightly regulated by transcriptional factors such as sterol regulatory element-binding factor (SREBF1), such as SREB protein 1A (SREB1A), 1C (SREBP1C) and SREBP2, liver X receptors (LXRs), X-box binding protein 1 (XBP-1), which also regulates endoplasmic reticulum (ER) stress. Proliferator-activated receptors (PPARs) that can either enhance genes involved in β-oxidation/translocation of FA (PPAR-α) or promote lipogenesis and glucose uptake and storage (PPAR-γ). The expression of these transcriptional modulators is altered in animal models and in patients with NAFLD/NASH [29].

The vast scientific literature suggests that phenolic phytochemicals may (1) interact with specific proteins in signaling pathways and modulate the activity and/or expression of key antioxidant proteins; (2) regulate the epigenetic mechanisms of gene expression; and (3) modulate the gut microbiota profile and metabolites. For example, flavone and tyrosol derivatives are implicated in the activation of AMP kinase (AMPK) and as a consequence in the regulation of several metabolic enzymes and autophagy that play a role in NAFLD [30,31]. AMPK activation has been shown to be mediated by direct interaction of certain polyphenols with phosphodiesterases [32], but it can be also modulated by NRH-quinone oxidoreductase 2 (NQO2) [30] and possibly by other enzymes among more than 5000 proteins predicted to be direct targets of these compounds [33].

The interaction of flavonoids with their protein targets leads to short- and long-term gene expression changes. The latter are mediated by epigenetic mechanisms such as DNA methylation, histone acetylation and deacetylation, causing spatial reorganization of the chromatin [34,35]. In fact, flavones and other dietary polyphenols have been shown to exhibit epigenetic modulatory effects on several gene targets [36,37], which are epigenetically altered in NAFLD [38,39].

Autophagy contributes to liver homeostasis through its role in cell quality control, by removing misfolded proteins, damaged organelles and lipid droplets [40]. Autophagy and energy metabolism gene expression programs are interconnected [41,42] and it is induced by flavonoids in many cell types including in vivo models of NAFLD [43,44]. For example, our group has demonstrated that BPF flavonoids stimulate evident bulk and lipid autophagy (lipophagy), detectable after few hours in cell culture [8,45] and upon chronic supplementation of BPF, efficiently preventing NAFLD in rats treated with CAF diet [9].

Here, we evaluated the impact of chronic supplementation of BPF in CAF-treated rats on mRNA levels of 84 genes representing different energy metabolism pathways, often altered in hepatic steatosis and 84 autophagy genes. Our data clearly show that suppression of de novo lipogenesis is a particularly potent effect of bergamot polyphenols at the level of gene expression, while β-oxidation and most autophagy genes are not subjected to transcriptional regulation in livers chronically treated with BPF.

## 2. Materials and Methods

### 2.1. Animal Procedures and Experimental Design

Male 5-week-old Rcc: Han WISTAR rats (Harlan Laboratories, Indianapolis, IN, USA) were housed two rats/cage in an animal housing facility, with access to water and standard chow (SC) diet 2016 (“SC”, Teklad, Harlan Lab.) or SC and CAF diet *ad libitum* and maintained in standard conditions as previously described [9]. CAF diet included different sweet or briny foods and condensed milk. The exact composition and feeding protocol have been previously described [9]. All animal studies were approved by the Italian Health Ministry and by the local ethics committee. At 8 weeks of age, the rats were weighed, marked on the tail for recognition, and randomly assigned to two experimental groups: CAF diet group (CAF, *n* = 10 rats) or SC diet group (SC, *n* = 5 rats). CAF group was subsequently subdivided into two subgroups, of which one received BPF extract (~50 mg/kg body weight/day) as a supplement in drinking water (CAF + BPF, *n* = 5) and the other received drinking water without BPF (CAF, *n* = 5). After a week of adaptation to the new cage mate, the administration of CAF diet started (day “0”) and lasted 91–94 days until the day of sacrifice. Food consumption and body weight gain were monitored weekly for 14 weeks. The animals were sacrificed under Zoletil (80 mg/kg) and Dormitor anesthesia for tissue collection.

The blood was collected by cardiac puncture as previously described [9] and cholesterol and triglycerides analyses were performed using commercial reagents on a Dimension EXL analyzer (Siemens Healthcare Diagnostics s.r.l., Milano, Italy).

### 2.2. Gene Expression Analysis on RT2-PCR Arrays

Three representing rats were chosen for each experimental group (SC, CAF, CAF + BPF). Small pieces (0.4–1 g) of the central part of the main lobe of rat livers were shock-frozen in liquid nitrogen and stored until needed at −80 °C. Frozen tissue was further fragmented and 50–100 mg samples were homogenized with a glass douncer on ice with 1 mL of TRIzol Reagent (Cat. No. 15596026, Invitrogen, Thermo Fisher Scientific, Waltham, MA, USA). Total RNA (totRNA) was extracted using the TRIzol Reagent method followed by DNase treatment (Cat. No. 79254, Qiagen Gmbh, Hilden, Germany). TotRNA was carefully quantified, and its integrity was verified on 0.8% denaturing agarose gel. Equal amounts of totRNA from 3 rats of the same experimental group were pooled and cDNA was synthesized by using RT2 PreAMP cDNA Synthesis Kit (Cat. No. 330451; Qiagen Gmbh) and 500 ng of pooled RNA for each RT2-PCR array, according to manufacturer instructions. The relative gene expression was assayed on 96-well format arrays: Rat Fatty liver (PARN-157Z) and Rat autophagy (PARN-084Z) RT2 Profiler PCR arrays (SABiosciences, Qiagen Gmbh), each containing 84 genes of interest, 6 housekeeping genes (hk) and control wells for genomic DNA (gDNA) contamination and RT-PCR efficiencies. RT-PCR was performed on an iQ5 real-time PCR (Bio-Rad Laboratoires, Inc., Hercules, CA, USA) using SYBR Green (universal cycling conditions: 95 °C, 10 min; 95 °C, 15 s; and 60 °C, 1 min; repeated for 40 cycles) and subsequent analyses were carried out according to the manufacturer’s recommended protocol (SABioscience, Qiagen Gmbh). Melt curve analysis confirmed the amplification of a single product. Three independent RT2-PCR assays with the control group pooled cDNA and two arrays for each CAF and CAF + BPF cDNA were performed. This and further analysis were performed using dedicated software available at the Gene Globe Data Analysis Center (http://www.qiagen.com/it/shop/genes-and-pathways/data-analysis-center-overview-page/ accessed on 3 April 2020). Briefly, the raw cycle threshold (CT) data of replicate plates for each experimental group (SC, CAF or CAF + BPF) were subjected to quality control, according to internal plate controls and standard parameters. Subsequently, two hk control genes—hypoxanthine phosphoribosyl transferase (*Hprt1*) and B2-microglobulin (*B2m*)—were selected for normalization and the software automatically calculated the normalized gene expression (2^−ΔCT^) for the Control (con) and Test Samples (ts), the fold change (2^−ΔΔCT^ = ts(2^−ΔCT^)/con(2^−ΔCT^)) and the fold regulation of gene expression based on the average ΔCT of replicate plates. To calculate the standard deviation (SD) of fold change in a gene analyzed on replicate plates, SD of ΔCT calculated by the software was added and subtracted from the average ΔCT to calculate maximum (max) and minimum (low) ΔCT. Subsequently max and low fold change (2^−ΔΔCT^) and the respective SD was calculated.

### 2.3. Quantitative (q)RT-PCR Analysis of Gene Expression on Individual RNA Samples

For standard qRT-PCR analysis, total (tot)RNA was isolated as described above from 5 rats for each experimental group, but each RNA was processed separately. cDNA was synthesized from 5 mg of totRNA with TransScript^®^ II First-Strand cDNA Synthesis SuperMix (Cat. No. AH301-02) according to manufacturer instructions (TransGen Biotech Co., Ltd., Haidian District, Beijing, China). qRT-PCR was performed on QuantStudio 3 Real-Time PCR Detection System (Applied Biosystems Europe, Monza (MI), Italy) as previously described [15]. The list of used primers is shown in Table 1, except for primers used for cytokine expression analysis: *Il10*, *Il1b*, *Il6*, *Infg* and *Tnf*, which were described previously [15]. The sequences of all primers were different from the oligos used for RT2-PCR array.

The applied cycling conditions were as follows: 95 °C, 10 min; 95 °C, 15 s; and 61 °C, 1 min; repeated for 40 cycles. Samples were analyzed in triplicate with hypoxanthine phosphoribosyl transferase 1 (*Hrpt1*) as a HK control. Each reaction was performed in triplicate for each individual cDNA and rat. Only results with the amplification of a single product, as verified by melting curve analysis, were considered. The mean CT of triplicate hk controls was subtracted from the mean of triplicate cycle thresholds (CT) of genes of interest to calculate ΔCT for each rat. Relative gene expression was calculated according to the formula: 2^ΔCT^.

### 2.4. Liver Histology and Lipid Droplets (LDs) Staining and Analysis

Frozen sections of the perfused liver, 10 μm thick, from the central portion of the main lobe, were prepared as previously described [9]. For lipid droplets (LDs) staining and analysis, please see the Appendix A.

### 2.5. Tissue Homogenization and Western Blotting (WB)

Liver fragments were homogenized on Bullet Blender Storm 24 tissue homogenizer (Next Advance, Inc., Averill Park, NY, USA) according to manufacturer indications, using 1 mm diameter Zirconium Oxide Beads (Next Advance. Inc.) at 4 °C in RIPA lysis buffer [Tris-HCl 20 mM (pH 7.5) NaCl 150 mM, Igepal 1%, EDTA 1 mM, SDS 0.1%] supplemented with protease inhibitors (cOmplete Mini, EDTA free, REF 11836170001; Roche Diagnostics Gmbh, Mannheim, Germany), NaF 2 mM and sodium orthovanadate 2 mM. Protein concentration was evaluated and then 40 µg was subjected to gradient 4–12% Bis-Tris electrophoresis NuPAGE (#NP0335BOX) according to manufacturer’s instructions (Invitrogen, Thermo Fisher Scientific) or 12% SDS-PAGE (for anti-LC3 and ATG16 WBs). WB was performed on polyacrylamide gel and the primary antibody was usually incubated overnight at 4 °C, followed by 1 h of incubation with a secondary antibody at RT. Blots were developed with ImmunoBlot ECL reagents (Cat. # 170-5061; Bio-Rad Lab., Inc.).

### 2.6. Antibodies

The antibodies used for WB were as follows: rabbit polyclonal (rp) anti-GCK (H-88) (sc-7908; Santa Cruz Biotechnology Inc., Dallas, TX, USA; 1:1000); rp anti-PCK2/PEPCK (sc-32879; Santa Cruz Biotech. Inc.; 1:400); rp anti-ACLY (Cat. No. 15421-1-AP; Proteintech Group, Inc., Rosemont, IL, USA; 1:1000); mouse monoclonal (mm) anti-ACACA/ACC (Cat. No: 67373-1-Ig, Proteintech Group, Inc.; 1:1000); rp anti-ATG16 (Code No. PM040Y; MBL International, Woburn, MA, USA; 1:1000); rp anti-ADRP/Perilipin 2 (Cat. No. 15294-1-AP, Proteintech Group, Inc.); mm anti-α-tubulin (T6074, Sigma Aldrich, Darmstadt, Germany; 1:1000); rp anti-LC3 (Code No. M186-3; MBL International; 1:1000); and rp anti-GAPDH (sc-87752, 1:500; Santa Cruz Biotech. Inc.; 1:500) were used as primary antibodies.

### 2.7. Data Analysis and Statistical Procedures

WB optical density was analyzed as previously described [46]. Each liver lysate (from 1 rat) was analyzed at least twice by WB, and the results were expressed as the mean ± standard error (SEM). The data were evaluated using ordinary one-way ANOVA followed by Tukey’s post-test and occasionally by uncorrected Fisher’s least significant difference (LSD) test, as indicated in the legends. Brown–Forsythe test followed by unpaired t with Welch’s correction was applied when significantly different standard deviations (SD) were found between groups. The differences were considered significant at *p* < 0.05.

## 3. Results

### 3.1. Bergamot Polyphenols Efficiently Prevent CAF Diet-Induced Hepatic Steatosis in Rats

CAF diet supplementation for 15 weeks according to Figure 1 induced typical NAFLD-related disorders in Wistar male rats. CAF diet-fed rats showed considerably higher body weight (Figure 2A), higher blood triglycerides (TGL) (Figure 2B), liver fat accumulation (Figure 2D) and a grade of micro- and macro-steatosis in the liver (Figure 3). The undesired effects of CAF diet were mitigated when it was supplemented with BPF (50 mg/kg/rat daily). Although the bioactive compounds did not exert a significant effect on final body weight (Figure 2A), they considerably reduced TGL and cholesterol levels in the blood (Figure 2B,C), as well as liver fat accumulation (Figure 2D). A potent reduction in hepatic steatosis by BPF was confirmed by a strong suppression of an LD-coating protein ADRP/Perilipin 2 in liver specimens (Figure 2E,F) as well as by oil red staining, which allows sensitive detection of intracellular TG and cholesterol esters.

This technique revealed the accumulation of numerous, considerably larger LDs in hepatocytes of CAF-fed rats (Figure 3B) compared to SC-fed rats (Figure 3A). BPF supplementation strongly attenuated hepatic LDs accumulation (Figure 3C). These observations were confirmed by the analysis of the numbers and size of lipid droplets (LD) in oil red stained liver sections (Figure 3G). Hematoxylin staining revealed important differences in liver histology: in SC and groups, liver parenchyma appeared homogeneous, and the hepatocytes showed uniform size with large, rounded nuclei usually located in the center of the cells and cytoplasmic glycogen granules, with few LDs (Figure 3D). In contrast, the cytoplasm of CAF hepatocytes appeared highly vacuolated, rich in both glycogen granules and LDs (Figure 3E). Importantly, CAF + BPF livers resembled SC with moderately increased oil red staining (Figure 3F).

### 3.2. Bergamot Polyphenols Strongly Suppress Lipogenesis-Related Genes in the Liver

To investigate the transcriptional changes induced by CAF diet and the chronic effects of BPF, we performed RNA profiling of key genes that might play a role in the pathogenesis of liver steatosis. The analysis was performed on pooled RNA samples from three representative rat livers for each experimental group. Pooling different RNAs has been validated in many studies as a reliable and cost-effective approach to reducing biological variability in differential expression analysis [47,48]. The RT2-PCR array analysis included a total of 84 genes (Figure 4C) belonging to the pathways of insulin and adipokine signaling, β-oxidation, cholesterol and lipid metabolism and transport, carbohydrate metabolism, inflammatory response and apoptosis. Compared to the control SC diet, the expression of 15 genes was considerably changed in the liver of rats fed with CAF diet (Figure 4A).

As expected for a diet rich in simple sugars, the CAF diet upregulated genes involved in glucose utilization (*Gck* and *Pck2*) and its conversion into lipids, although lipogenesis genes *(Acly*, *Acaca* and *Fasn*) were only moderately induced in the RT2 array experiment. In the CAF vs. SC group (Figure 4B) 10 genes related to lipid transport (*Fabp3*, *Lpl*), carbohydrate metabolism (*Gck*, *Pck2*), endoplasmic reticulum (ER) stress (*Xbp1),* inflammatory response (*Il6*), and cholesterol metabolism and transport (*Apoa1*, *Lepr*, *Pparg*, *Srebf1*) were upregulated, and 5 genes related to adipokine signaling (*Cd36* and *Serpine1*), beta oxidation (*Cpt1a*), insulin signaling (*Insr*), and lipid metabolism (*Hnf4a*) were downregulated. Interestingly in CAF + BPF vs. CAF experimental groups (Figure 4D), 23 genes related to cholesterol and lipid metabolism (*Abcg1*, *Apoa1*, *Ldlr*, *Lepr*, *Nr1h3*, *Srebf2*, *Acaca*, *Fabp5*, *Fasn*, *Lpl*, *Scd*), insulin and adipokine signaling (*Akt1*, *Slc2a1*, *Slc2a4*, *Ppargc1a*), carbohydrate metabolism (*Acly*, *G6pd*, *Gck*, *Pck2*), inflammatory response (*Il10*, *Il6*, *Tnf)* were downregulated and surprisingly *Cyp2e1*, presumably related to fatty liver phenotype, was upregulated. For the gene expression analysis between CAF vs. SC (Figure 4B) and CAF vs. CAF + BPF (Figure 4E) groups, a gene induction higher than 1.7-fold and a reduction in expression to at least −1.7 were defined as the cut-off values. The majority of genes were regulated below the threshold and they were equally distributed between positive and negative fold regulation (Figure 4C,F). 

### 3.3. BPF Supplementation Has a Minor Effect on Liver Autophagy Gene Expression

Next, mRNA levels of autophagy-related genes were assessed in the same samples of cDNA, but on autophagy-specific arrays. A scatter plot of 84 autophagy-related genes, showed that the expression of 5 genes *(Tnf*, *Ins2*, *Atg16I2*, *Map* and *Hsp90*) were significantly modulated in CAF vs. SC group (Figure 5A,B) while comparison between the CAF + BPF and CAF group showed changes in only 3 genes (*Ins2*, *Map1lc3a* and *Dapk1*) (Figure 5C,D). Thus, compared to the genes responsible for the development of fatty liver, autophagy-related gene expression is much less affected by bergamot polyphenols. The data presented in the scatter plots in Figure 4A,D and Figure 5A,C were also normalized to the expression levels in SC group (Figure 6).

### 3.4. Standard qRT-PCR Confirms RT2-PCR Array Data on BPF-Induced Gene Modulation

Differential expression of a subset of candidate regulated genes identified by RT2-PCR array approach was confirmed by individual qRT-PCR assays. This approach was useful to assess the biological variability lost by pooling RNAs, and to compensate for the limited number of technical replicas in RT2-PCR array analysis. The induction of lipogenesis genes in CAF rats was statistically significant for *Srebf1* and *Pparg*, but not for *Fasn*, *Acly* and *Acaca*, when mRNA from four to five individual animals was separately analyzed (Figure 7). In addition, the CAF diet caused a dramatic upregulation in *Gck*, *Pck2*, *Ins* genes and a significant increase in *Il6*, *Il10*, *Tnf* and *Il1b* compared to the SC group without affecting *Srebf2*, *Ppara* and *Infg* genes (Figure 7), which showed a similar trend as observed in array analysis (Figure 6). In line with the array data, a marked downregulation in the majority of lipogenesis- and diabetes-related genes (*Pck2*, *Gck* and *Ins*) as well as in cytokines was observed in BPF-treated livers compared to both CAF and SC groups (Figure 7). Interestingly, *Fasn*, *Acaca*, *Acly*, *Pck2* and *Ins* were suppressed far below the mRNA levels of the control livers. Finally, the CAF diet significantly reduced *Cpt1b* expression and it was not upregulated in CAF + BPF livers, while another β-oxidation gene *Ppara* was unchanged (Figure 7). All the data presented as fold change in Figure 7 were also shown as *Hrpt* relative expression (Appendix A, Appendix A).

### 3.5. Most BPF-Induced Effects on Gene Expression Can Be Replicated at the Protein Level

To verify the BPF-induced effects on gene expression, we performed a Western blot analysis of liver lysates for the most representative gene products. The increased ACACA and PCK2 protein levels in rats fed with CAF diet were reversed by BPF supplementation and in the case of ACACA were significantly reduced when compared with the SC group (Figure 8A,B). The CAF diet was not able to modulate the protein levels of ACLY and GCK enzymes in comparison to the SC diet, but BPF treatment significantly decreased it (Figure 8A,B). This finding contrasts with the highly elevated *Gck* mRNA in CAF livers (Figure 4B,C, Figure 6A and Figure 7), suggesting that not all significant transcriptional effects are mirrored by protein levels. Concerning the expression level of autophagy-related proteins, both LC3I and LC3II forms of LC3 were significantly reduced under CAF treatment when compared to the SC group (Figure 8C,D) and only autophagosome marker LC3II was significantly upregulated in the CAF + BPF group with respect to the CAF group. However, in contrast to RNA expression data in Figure 5, ATG16L was found to be slightly upregulated in the CAF group but not modulated in the CAF+ BPF group (Figure 8C,D).

## 4. Discussion

*Citrus bergamia* flavonoids efficiently prevent NAFLD, systemic redox imbalance and other features of metabolic syndrome in rats fed with CAF diet [9,49]. These original findings have been subsequently confirmed in other models and extended to NASH [13,15,50] and NAFLD patients with metabolic syndrome patients in subsequent studies [17,18]. Accordingly, the rats used in the present study responded to BPF supplementation with a strong reduction in lipid accumulation in CAF-fed livers characterized by a 60% and 80% decrease in fat content and numbers of big lipid droplets, respectively, confirmed by histopathological analysis of liver sections. BPF largely and significantly reduced hypertriglyceridemia in CAF rats, while it had no significative impact on the body mass in this experiment; this confirms that BPF is hepatoprotective but has no or little effect on obesity [9,15]. In this article, we evaluated the impact of chronic supplementation of BPF on mRNA levels of a battery of 168 genes to address possible mechanisms behind the widely demonstrated efficacy of bergamot polyphenols against NAFLD.

Although some RT2-PCR array data presented in Figure 4, Figure 5 and Figure 6 should be interpreted with caution, due to a limited number of technical replicates and no independent qRT-PCRs data, several interesting observations can be made. We can state that the gene expression analysis revealed that the CAF diet (14 weeks) produced both expected and few unexpected effects on the transcriptional profiles of hepatic genes compared to other hypercaloric diets. There is a pattern of insulin and leptin resistance characterized by upregulation of insulin mRNA by 5 to 10 times, downregulation of *Insr* mRNA but upregulation of leptin receptor *Lepr* transcript by 2- and 2.3-fold, respectively. The key transcription factors of lipid and cholesterol synthesis *Srebf1*, *Xbp1* and *Pparg*, were also upregulated around 2-fold, which is common to many rodent models of NAFLD and found in patients with histologically diagnosed NAFLD [29].

However, there is no change or not statistically significant increase in transcripts coding for lipogenesis enzymes (*Acly*, *Acaca*, *Fasn*, *Scd-1*) in CAF livers. This is in contrast to fructose-induced rat and murine models of NAFLD, where the main targets of SREBP-1c-mediated transactivation, *Acaca* and *Fasn*, were upregulated 6- to 16-fold compared to a control animal [22,51]. Yet, such a strong difference between those models might be explained by overnight fasting with continued fructose supply before sacrifice [52], while in our study all the animals were deprived for 5 h of all energy sources for blood analysis. Indeed, a study reported that in high-fat diet (HFD)-fed mice, *Acaca* was more expressed in HFD than in the control group and decreased to baseline levels in fasted control animals. In the same experiment, *Fasn* upregulation by 3- to 4-fold in HFD mice was less dependent on fasting [23]. Accordingly, we found consistent, but modest upregulation of *Fasn* mRNA in the CAF group by around 1.6, but less consistent data for *Acaca* and *Acly* in both array pools (Figure 4) and individual assays (Figure 7), suggesting that lipogenesis enzyme expression is flexible and quickly responds to nutrient status. In contrast, the upstream transcription factors, such as *Srebf1* and *Pparg* maintained the stable 2-fold upregulation, regardless of short-term fasting [23].

According to the phenotypic changes in rat livers, BPF was able to suppress several NAFLD-related genes. Among them, mRNA levels of *Acly*, *Acaca*, *Fasn*, *and Scd1* were strongly suppressed in BPF-treated livers far below mRNA levels of CAF and control livers, as confirmed also by an alternative primer-pair qRT-PCR analysis. For some genes, we could also observe that their mRNA downregulation (*Acaca* and *Acly*) led to a marked protein decrease in BPF-treated livers. BPF also drastically downregulated liver expression of *Ins2*, suggesting a powerful improvement in insulin sensitivity in CAF + BPF rats and confirming our previous observations in rats exposed to CAF diet and then to BPF and SC diet [15]. Hyperinsulinemia potentiates lipogenesis and FA accumulation leading to hepatotoxicity and inflammation/fibrosis, while insulin sensitivity prevents these effects by favoring the transport of fat from liver to adipose stores. Extrahepatic lipogenesis and adipogenesis serve as compensatory agents for improving insulin sensitivity and protecting the liver from fat accumulation [53]. This might explain why BPF exerts potent hepatoprotective effects but has little or no effect on obesity in the presence of hypercaloric diet.

The therapeutic potential of different polyphenols and plant extracts, including Citrus fruit extracts, have been tested in several NAFLD and NASH rodent models, but none of these studies reported such a dramatic transcriptional suppression of lipogenesis as bergamot polyphenols in CAF-induced NAFLD/NASH in rats. This is true for both purified compounds as well as for complex plant extracts [44,52,54,55,56,57,58]. In another study, the same array of 84 mouse genes related to NAFLD was used to characterize the antisteatotic effect of a complex mixture of natural extracts containing silymarin, curcumin and chlorogenic acid. Yet, even though this extract almost fully reverted NAFLD induced by HFD in mice, only *Scd-1* and *Fabp5* were found to be downregulated after 16 weeks of treatment [55].

Another important finding in this work is no regulation of FA oxidation genes, such as *Acadl*, *Acox1*, *Cpt1a*, *Cpt2*, *Fabp1*, *Irs1*, *mTOR*, and *Ppara* by long-term BPF supplementation. The only effect was lower *Cpt1a* expression in CAF and CAF + BPF livers with respect to the SC group (Figure 7). CPT1A is the key enzyme in the carnitine-dependent transport of fatty acids across the external mitochondrial membrane. Its lower expression should limit fatty acids uptake by mitochondria and thus reduce β-oxidation, as an adaptative process to mitochondria-derived oxidative stress, found in fructose- [26,27], but not in high-fat diet-induced NAFLD models [27,59]. In fact, both mitochondrial as well as peroxisomal fatty acid oxidation are ROS-generating pathways and their excessive stimulation is deleterious for liver tissue [21]. Lack of positive stimulation of FA oxidation gene expression in BPF-treated livers is in line with a clinical study demonstrating that a multifactorial diet (rich in polyphenols and polyunsaturated fatty acids) downregulated lipogenesis, but did not regulate plasma levels of a β-oxidation marker, β-hydroxybutyrate [60]. Remarkably, the stimulation of hepatic β-oxidation is a typical response to different phytochemicals in vitro in cellular models of NAFLD [61,62,63,64], but it is only occasionally reported in rodents treated with purified phenolic compounds [65,66]. In conclusion, our data demonstrate that transcriptional suppression of lipogenesis, and not stimulation of β-oxidation, is the main mechanism of BPF counteracting fat accumulation in NAFLD and NASH in vivo as depicted in Figure 9. This finding is consistent with an antioxidant function of flavonoids, mediated by the regulation of expression of genes contributing to healthy redox balance rather than through direct ROS scavenging.

An unexpected effect of BPF was the induction of Cytochrome 2e1 (*Cyp2e1*) by BPF, a phase I drug metabolism enzyme, but not by CAF diet feeding without BPF. At first glance, it seems surprising, since CYP2E1 activity and expression were found to be elevated in human steatohepatitis and rodent alcohol and methionine-induced liver steatosis [67]. However, the increase in *Cyp2e1* expression has not been clearly established in rodent NAFLD models [67]. Interestingly, CYP2E1 is reversibly inhibited by flavonoids, while it contributes to flavonoid phase I metabolism in the liver [68]. Thus, its induction in our model might be a compensatory response to BPF flavonoids. In line with our findings, mice treated with flavone-8-acetic acid showed a substantial induction of hepatic CYP2E1 [69].

Modulation of autophagy to degrade LDs and relieve hepatic inflammation is a potential therapeutic target for NAFLD. Since common transcription factors are believed to regulate lipid metabolism and autophagy [41,70], we expected huge transcriptional changes in hepatic autophagy genes, induced by both the CAF diet and bergamot polyphenols. Surprisingly, we found that both the CAF diet and BPF had limited effects on autophagy-related gene expression. Indeed, the CAF diet slightly upregulated *Hsp90aa1* and downregulated *Atg16l2*, and *Maplc3a*, while BPF only upregulated *Maplc3a* and *Dapk1*. The other two genes, *Ins2* and *Tnf*, upregulated by the CAF diet and downregulated by BPF, are not typical autophagy-related genes and they play more important roles in diabetes and inflammation. At the protein level, we were able to confirm the downregulation of LC3I and LC3II levels in CAF livers and its upregulation by BPF diet, but not in the case of ATG16L, suggesting that there is a partial correlation between mRNA levels and protein expression. This indicates that polyphenols regulate autophagy mainly by post-translational mechanisms. In fact, Beclin-1 and p62/SQSTM1 were previously found to be regulated by chronic supplementation of BPF in rats [9], but they are not regulated at mRNA levels in this study. To our knowledge, the dataset presented here is the first characterization of autophagy-related gene expression in response to the CAF diet and polyphenols.

## 5. Conclusions

In conclusion, among the many pleiotropic effects of polyphenols, the main mechanisms underlying the anti-steatotic effect of BPF supplementation appear to be transcriptional and include a potent suppression of lipogenesis and an effective reduction in gluconeogenesis. Interestingly, β-oxidation is not induced transcriptionally, while autophagy-related genes are only marginally modulated by chronic treatment with BPF, indicating that autophagy is regulated mainly by posttranslational mechanisms.

Considering that oxidation of fatty acids is the main source of oxidative stress in NAFLD, no induction of β-oxidation with an extremely potent suppression of lipogenesis might be an efficient antioxidant mechanism of bergamot flavonoids in vivo. This mechanism provides a convincing explanation of the high efficacy of BPF against NAFLD, far beyond its anti-inflammatory and proautophagic effects.

## Figures and Tables

**Figure 1 antioxidants-13-00766-f001:**
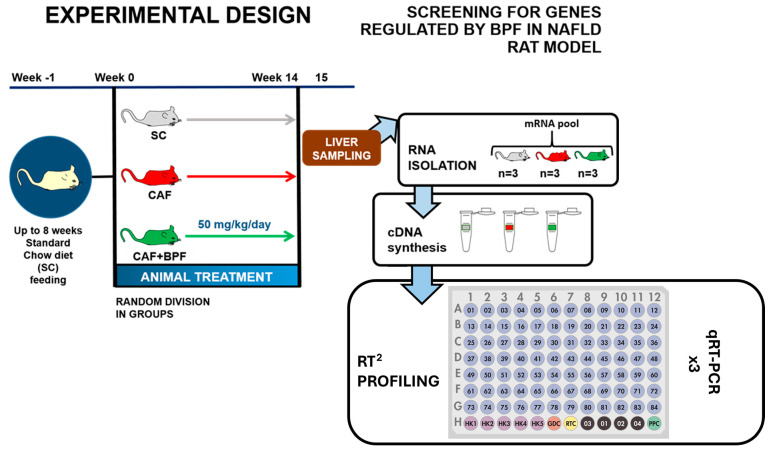
The experimental design and screening for genes regulated by BPF in NAFLD rat model in this work: rat division for dietetic treatment and BPF administration for 15 weeks. On the right side, a flow description is exhibited, from the liver sampling to RT2 profiling.

**Figure 2 antioxidants-13-00766-f002:**
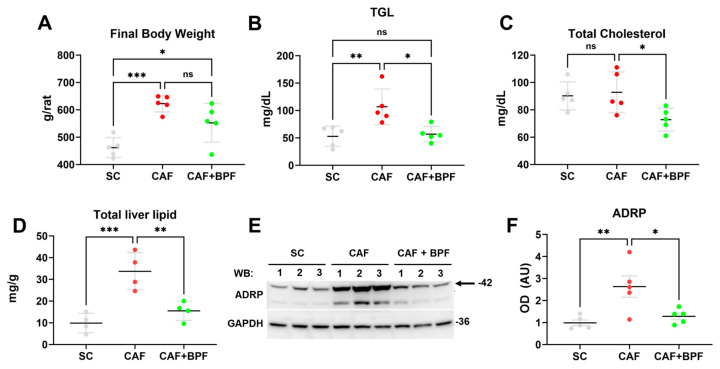
Bergamot polyphenols prevent CAF diet-induced obesity, hypertriglyceridemia and intracellular fat accumulation in Wistar rats. (**A**) Final body weight, (**B**) blood triglycerides, (TGL) (**C**) blood total cholesterol. Data are presented as the mean ± SD of *n* = 5 rats. Each dot represents a rat. (**D**) The total lipid content in 400 mg of liver tissue was determined by Folch’s method. Data are presented as the mean ± SD of *n* = 4 livers for each group. (**E**) Representative blots for ADRP/Perilipin 2 and GAPDH as a loading control, showing liver lysates from 3 different rats for each group. (**F**) OD ratio of ADRP to GAPDH expression levels. Data are expressed as the mean ± SEM of *n* = 6 rat liver lysates for each group. Statistical analysis in (**A**–**D**,**F**): one-way ANOVA with Tukey’s post-test, * *p* ≤ 0.05, ** *p* ≤ 0.01, *** *p* ≤ 0.001, ns—not significant change.

**Figure 3 antioxidants-13-00766-f003:**
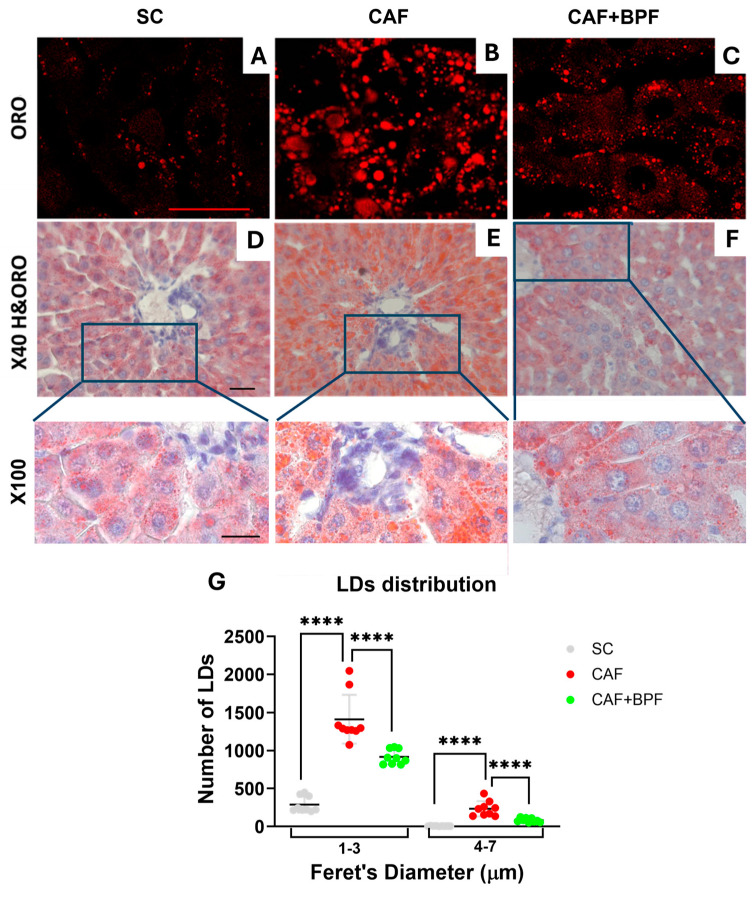
BPF prevents CAF diet-induced hepatic steatosis. Histopathological changes of rat liver tissues between different dietary groups. (**A**–**C**) Representative hematoxylin (H) and oil red (ORO) stained liver sections were visualized by confocal microscopy, bar = 25 μm; and (**D**–**F**) by bright-field, magnification ×40, bar = 40 μm. Images in the third row show magnified regions of D to F images indicated by blue boxes, bar = 20 μm. (**G**) LDs size and number quantification on confocal sections (as in **A**–**C**). LDs between 1 and 3 µm indicate microsteatosis and 4–7 µm macrosteatosis, respectively. Data are presented as the mean ± SEM (*n* = 3 livers and 9 images for each group) for LDs between 1 and 3 µm and 4–7 µm, respectively. Statistical analysis: One-way ANOVA with Tukey’s post-test. **** *p* ≤ 0.0001.

**Figure 4 antioxidants-13-00766-f004:**
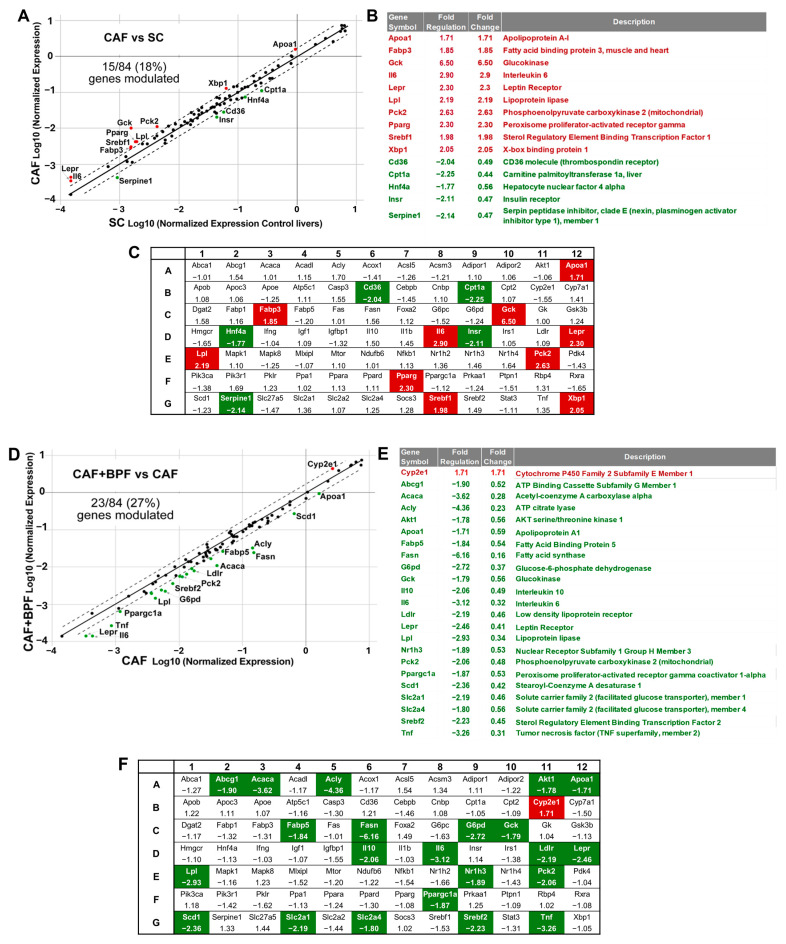
Differential expression of fatty liver-related transcripts in steatotic livers exposed to CAF diet for 14 weeks as compared to control SC diet (**A**–**C**) and the effects of a chronic supplementation of bergamot polyphenols to CAF diet in rats (**D**–**F**). (**A**) CAF vs. SC and (**D**) CAF + BPF vs. CAF scatter plots for differential expression analysis of 84 fatty liver genes. Below, the lists of genes modulated more than 1.7-fold when CAF livers are compared to SC livers (**B**) and CAF + BPF (**E**) are compared to CAF livers. (**B**,**D**) The arrays of all fatty liver-associated genes with respective fold regulation values when (**C**) CAF vs. SC and (**F**) CAF + BPF vs. CAF livers are compared. Note that bergamot polyphenols strongly suppress lipogenesis-related genes in the liver.

**Figure 5 antioxidants-13-00766-f005:**
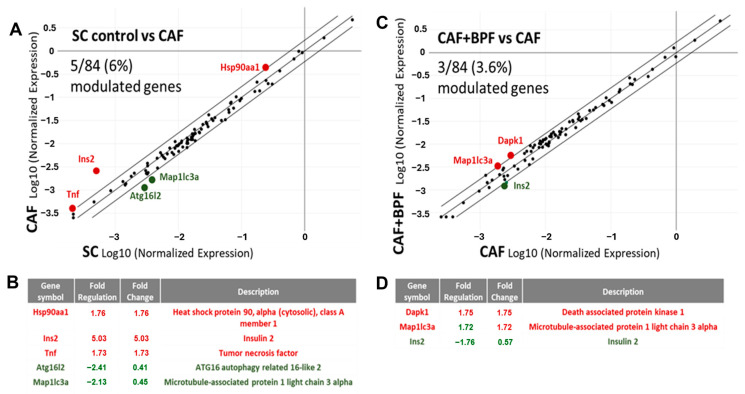
Differential expression of autophagy-related transcripts in steatotic livers exposed to CAF diet for 14 weeks as compared to control SC diet (**A**,**B**) and the effects of chronic supplementation of bergamot polyphenols to CAF diet in rats (**C**,**D**). (**A**) CAF vs. SC and (**C**) CAF + BPF vs. CAF scatter plots for differential analysis of 84 autophagy genes. (**B**,**D**) Below, the lists of genes modulated more than 1.7-fold when CAF livers are compared to SC livers (**B**) and CAF + BPF are compared to CAF livers (**D**). Note that CAF diet and bergamot polyphenols have a very modest effect on the expression of autophagy-related genes.

**Figure 6 antioxidants-13-00766-f006:**
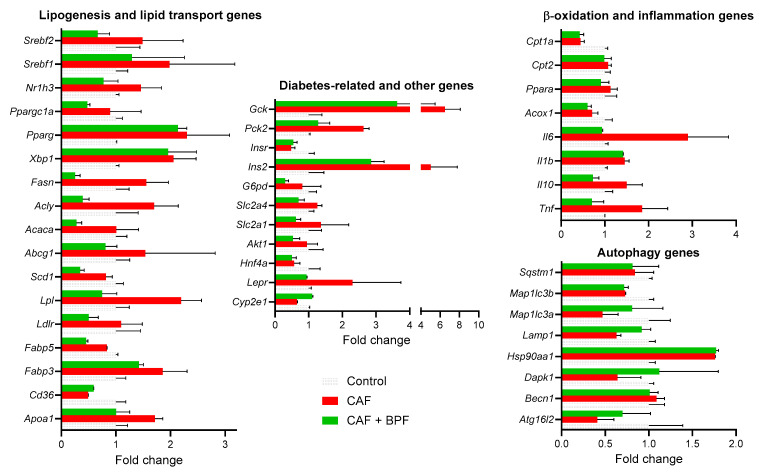
Differentially expressed hepatic genes in CAF and CAF + BPF groups when compared to control SC group. Fold change expression of selected lipogenesis and lipid transport (**left**), diabetes-related (**center**), β-oxidation, inflammation and autophagy-related transcripts (**right**) in CAF diet- and CAF + BPF-treated livers normalized to control (SC) livers. Data are presented as mean ± SD. See Section 2.

**Figure 7 antioxidants-13-00766-f007:**
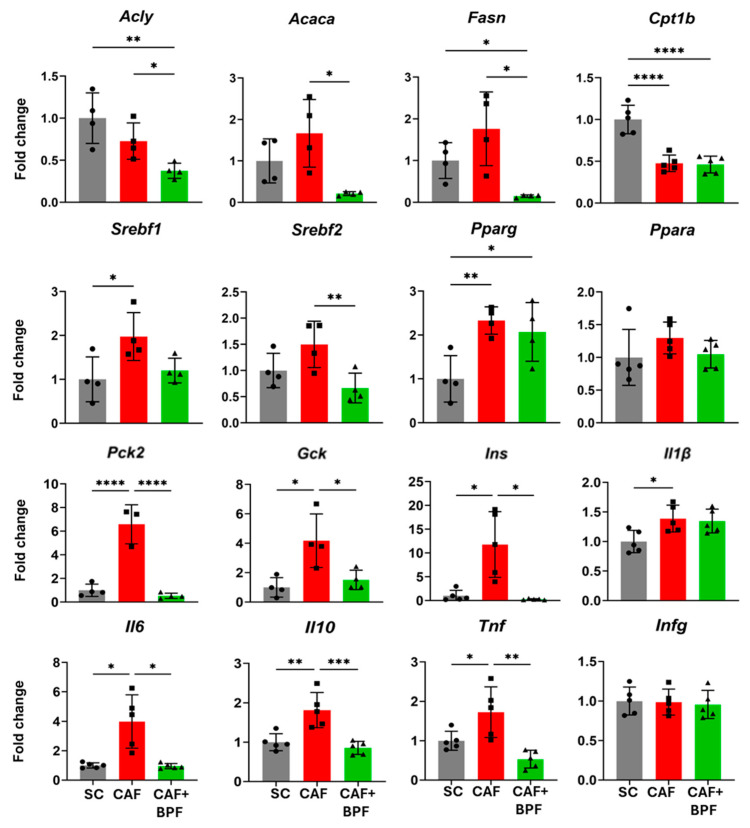
The expression level of selected genes was analyzed by qRT-PCR with an independent set of primers for each animal separately. The bars represent fold change expression from *n* = 4 to 5 rats in CAF diet- and CAF + BPF-treated livers normalized to control (SC) livers. Data are presented as mean ± SD. Statistical analysis: one-way ANOVA followed by Tukey’s post-test or uncorrected Fisher’s LSD test, except for *Acaca*, *Fasn*, *Ins* and *Il6* in which Brown–Forsythe test followed by unpaired *t* test with Welch’s correction was applied. * *p* ≤ 0.05, ** *p* ≤ 0.01, *** *p* ≤ 0.001, **** *p* ≤ 0.0001.

**Figure 8 antioxidants-13-00766-f008:**
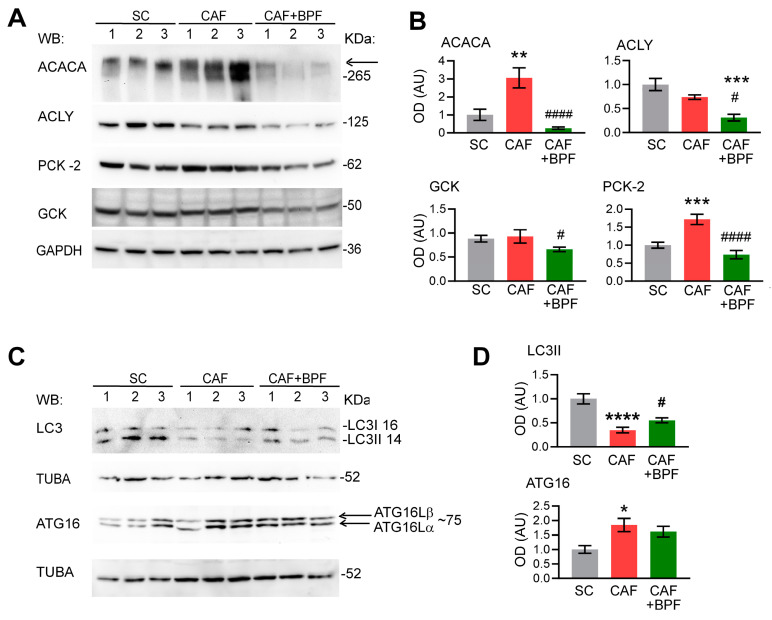
Western blot analysis of protein products of selected genes modulated by CAF diet and BPF. (**A**) Representative blots showing 3 rat livers, for each group, for ACACA, ACLY, PCK2 and GCK. ADRP/Perilipin 2 has been shown as a marker for steatosis or lipid content and GAPDH as a loading control. (**B**) OD analysis of expression levels of proteins as in A compared to GAPDH *n* = 5 to 6 rat livers for each group. (**C**) Representative blots showing protein lysates from 3 rat livers for each group for autophagy proteins LC3B and ATG16. Alpha-tubulin (TUBA) was used as loading control. (**D**) OD ratio of proteins as in C compared to TUBA in *n* = 5 to 6 rat liver samples for each group. Bars show the mean OD ratio ± SEM, normalized to the mean of SC group. Statistical analysis: one-way ANOVA with Tukey’s post-test or with LSD Fisher test for GCK OD analysis. *, **, ***, **** significant difference compared with control SC group at *p* < 0.5, *p* < 0.01, *p* < 0.001 or *p* < 0.0001, respectively. #, #### significant difference compared with CAF group at *p* < 0.5 or *p* < 0.0001, respectively. Numbers on the right of blots indicate the approximate position of molecular weights expressed in kDa.

**Figure 9 antioxidants-13-00766-f009:**
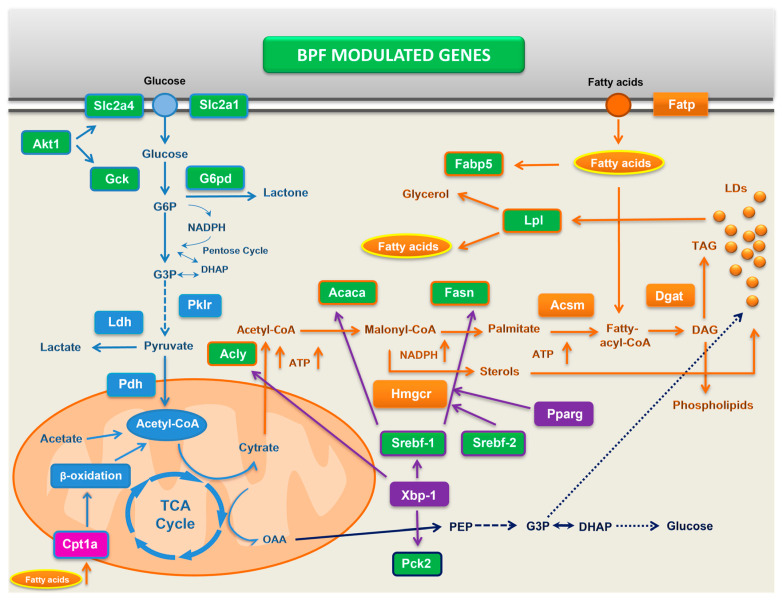
Schematic representation of main lipid- and glucose-metabolism genes differentially expressed in CAF + BPF-treated livers. Green rectangles: genes downregulated by BPF; pink rectangles: genes downregulated by CAF diet; light-blue rectangles and outlines: genes related to glucose metabolism; orange rectangles and outlines: genes related to lipid metabolism; violet rectangles and outlines: transcription factors coding genes; blue: intermediaries in gluconeogenesis pathway. Continuous and dotted arrows: direct and indirect connections, respectively.

**Table 1 antioxidants-13-00766-t001:** List of rat-specific primers for qRT-PCR analysis of gene expression. F—forward, R—reverse, bs—bases.

Accession	Gene Name	Primer	Sequence 5′-3′	No. bs
NM_019130.2	*Ins2*	INS2-F	ATC AGC AAG CAG GTC ATT GTT CCA	24
	Rattus norvegicus insulin 2	INS2-R	CTT CGC GGC GGG ACA TGG	18
NM_016987.2	*Acly*	ACLY-F	CGG CTC ACA CTG CCA ACT TC	20
	ATP citrate lyase	ACLY-R	TGG GAC TGA ATC TTG GGG CA	20
NM_022193.1	*Acaca*	ACC1-F	CTT CGG GGT GGT TCT TGG GT	20
	acetyl-CoA carboxylase alpha	ACC1-R	TTC CAG AAC GGA TCC CCT GC	20
NM_017332.1	*Fasn*	FASN-F	ATT GTG GGC GGG ATC AAC CT	20
	fatty acid synthase	FASN-R	CGG CAA TAC CCG TTC CCT GA	20
NM_013200	*Cpt1b*	CPT1B-F	GTT ATC GAG TTC AGA AAC GAA CGC	24
	Carnitine palmitoyltransferase 1B	CPT1B-R	CAC CCC TTA TGC CTG TGA ACT	21
NM_013196.2	*Ppara*—peroxisome proliferator	PPARA-F	AAT CCA CGA AGC CTA CCT GA	20
	-activated receptor alpha	PPARA-R	GTC TTC TCA GCC ATG CAC AA	20
NM_013124.3	*Pparg*—peroxisome proliferator	PPARG-F	AGC ATG GTG CCT TCG CTG AT	20
	-activated receptor gamma	PPARG-R	GCC CAA ACC TGA TGG CAT TGT	21
NM_012565.2	*Gck*	GCK-F	AGG TGT GGA GCC CAG TTG TTG	21
	glucokinase	GCK-R	TCC GAC TTC TGA GCC TTC TGG G	22
NM_001108377.2	*Pck2*	PCK2-F	GGT TGA GCA TGG AGG GAC GA	20
	phosphoenolpyruvate carboxykinase 2	PCK2-R	CTA GCA CGC GAG CGT TTT CC	20
NM_001276707.1	*Srebf1*—sterol regulatory element	SREBF1-F	CTC TTG ACC GAC ATC GAA GAC AT	23
	binding transcription factor 1	SREBF1-R	CCC AGC ATA GGG GGC ATC AA	20
NM_001033694.1	*Srebf2*—sterol regulatory element	SREBF2-F	GGC TGT CGG GTG TCA TGG G	19
	binding transcription factor 2	SREBF2-R	CTG TAG CAT CTC GTC GAT GTC C	22
NM_012583.2	*Hprt1*—hypoxanthine	HPRT-F	CTC ATG GAC TGA TTA TGGACAGGAC	25
	phosphoribosyltransferase 1	HPRT-R	GCAGGTCAGCAAAGAACTTATAGCC	25

## Data Availability

The data are contained within the article and Appendix A. Other original data supporting reported results are available upon request.

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
