# Peer review of "Dramatic Suppression of Lipogenesis and No Increase in Beta-Oxidation Gene Expression Are among the Key Effects of Bergamot Flavonoids in Fatty Liver Disease"

_antioxidants, 2024, doi:10.3390/antiox13070766_

Round 1

Reviewer 1 Report

The manuscript of Parafati et al. deals with Bergamot flavonoids that seem to prevent metabolic syndrome, non-alcoholic fatty liver disease (NAFLD) and stimulate autophagy in animal models and patients. Rats received for 14 weeks standard, “cafeteria” (CAF) and CAF supplemented with 50 mg/kg of Bergamot polyphenol fraction (BPF) diets. CAF diet caused a strong upregulation of the gluconeogenesis pathway, a moderate (>1.7 fold) induction of genes regulating lipogenesis (Srebf1, Pparg, Xbp1), lipid and cholesterol transport or lipolysis (Fabp3, Apoa1, Lpl) and inflammation (Il6), while one b-oxidation gene (Cpt1a) and a few autophagy genes were modulated in the CAF group. Most of these transcripts were significantly modulated by BPF, particularly with a potent effect on lipogenesis genes, like Acly, Acaca, Fasn and Scd1, which were suppressed far below the mRNA levels of control livers. Therefore, authors conclude that chronic BPF supplementation efficiently prevents NAFLD by modulating hepatic energy metabolism and inflammation gene expression programs, leading to a profound suppression of de-novo lipogenesis. Present data could be useful in the implementation of a therapeutical approach for NAFLD; however, I have some concerns for this study.

 1)    Authors say that the BPF diet containing bioactive compounds did not exert a significant effect on final body weight, but reduced blood TGL and cholesterol levels. If the flavonoid fraction has potent and coordinated changes in lipid metabolism, why did not affected body weight?

 2)    The CAF diet produces an evident liver fat accumulation in a grade of micro and macro-steatosis; however, the induction of lipogenesis genes, Fasn, Acly and Acaca, was not statistically significant. How authors explain this?

 3)    Specifically, what would be the source of malonyl-CoA and free fatty acids if the enzymes activities involved in lipogenesis are not increased?  

 4)    It is established that BPF is hepatoprotective but has no or little effect on obesity. Why specific mechanism allowed by the flavonoids fraction could provide specificity for the damaged liver tissue?

 5)    Authors say that BPF induced the Cytochrome 2e1 (CYP2E1), but not by CAF diet feeding without BPF. Then, it is possible that CYP2E1 could play a role in polyphenol metabolism, and its induction is a hepatic response to flavonoids and related metabolites. In fact, what’s known about the pharmacodynamic of BPF? The metabolism of BPF is mainly hepatic? In the case of the Alcoholic Fatty Liver Disease, it is possible to find a catabolic competition between alcohol and BPF?

The manuscript of Parafati et al. deals with Bergamot flavonoids that seem to prevent metabolic syndrome, non-alcoholic fatty liver disease (NAFLD) and stimulate autophagy in animal models and patients. Rats received for 14 weeks standard, “cafeteria” (CAF) and CAF supplemented with 50 mg/kg of Bergamot polyphenol fraction (BPF) diets. CAF diet caused a strong upregulation of the gluconeogenesis pathway, a moderate (>1.7 fold) induction of genes regulating lipogenesis (Srebf1, Pparg, Xbp1), lipid and cholesterol transport or lipolysis (Fabp3, Apoa1, Lpl) and inflammation (Il6), while one b-oxidation gene (Cpt1a) and a few autophagy genes were modulated in the CAF group. Most of these transcripts were significantly modulated by BPF, particularly with a potent effect on lipogenesis genes, like Acly, Acaca, Fasn and Scd1, which were suppressed far below the mRNA levels of control livers. Therefore, authors conclude that chronic BPF supplementation efficiently prevents NAFLD by modulating hepatic energy metabolism and inflammation gene expression programs, leading to a profound suppression of de-novo lipogenesis. Present data could be useful in the implementation of a therapeutical approach for NAFLD; however, I have some concerns for this study.

 1)    Authors say that the BPF diet containing bioactive compounds did not exert a significant effect on final body weight, but reduced blood TGL and cholesterol levels. If the flavonoid fraction has potent and coordinated changes in lipid metabolism, why did not affected body weight?

 2)    The CAF diet produces an evident liver fat accumulation in a grade of micro and macro-steatosis; however, the induction of lipogenesis genes, Fasn, Acly and Acaca, was not statistically significant. How authors explain this?

 3)    Specifically, what would be the source of malonyl-CoA and free fatty acids if the enzymes activities involved in lipogenesis are not increased?  

 4)    It is established that BPF is hepatoprotective but has no or little effect on obesity. Why specific mechanism allowed by the flavonoids fraction could provide specificity for the damaged liver tissue?

 5)    Authors say that BPF induced the Cytochrome 2e1 (CYP2E1), but not by CAF diet feeding without BPF. Then, it is possible that CYP2E1 could play a role in polyphenol metabolism, and its induction is a hepatic response to flavonoids and related metabolites. In fact, what’s known about the pharmacodynamic of BPF? The metabolism of BPF is mainly hepatic? In the case of the Alcoholic Fatty Liver Disease, it is possible to find a catabolic competition between alcohol and BPF?

Author Response

  • Rebuttal letter to reviewers’ comments on Ms ID: antioxidants-2976075

    We thank the editor and reviewers for their constructive comments that helped us to improve the quality of our manuscript. We addressed all comments and performed new experiments (qRT-PCR analysis for other 7 genes) with the aim to verify our findings according to reviewers’ suggestions. In particular, we explained the rationale behind pooling the RNA samples for RT2-PCR arrays and supported by a vast literature on the topic. In addition, we made changes in the text and in the figures (2, 3, 4, 6, 7 and 8) as requested by the reviewers to further improve the clarity and data presentation. Supplementary Information file was also improved. Finally, we modified slightly the title to respond, at least in part to reviewers’ request.

    For clarity we accepted all the changes in the revised final version of our manuscript Ms (MsR1) but provide an additional version where all the changes have been marked automatically by the Microsoft Office Word tracking system.

    Pointo to point reply to Reviewer 1 comments:
  •  
  • Authors say that the BPF diet containing bioactive compounds did not exert a significant effect on final body weight, but reduced blood TGL and cholesterol levels. If the flavonoid fraction has potent and coordinated changes in lipid metabolism, why did not affected body weight?

This is an interesting issue and the answer might be related to distinct mechanisms regulating hepatic and adipose lipogenesis and depending on insulin activity. In particular, peripheral insulin resistance and high glucose levels in the blood promote the vicious spiral of systemic hyperinsulinemia, which accelerates lipogenesis in the liver. Hyperinsulinemia-induced lipogenesis and FA accumulation cause hepatotoxicity and inflammation/fibrosis. As the largest and safest storage site for excess fats, subcutaneous de novo lipogenesis and adipogenesis serve as compensatory agents for improving insulin sensitivity by producing insulin-sensitizing fatty acid palmitoleate and protecting the liver from fat accumulation (Wu et al., 2023). BPF may exerts an effect on these pathways since it strongly downregulates insulin expression (as shown in this Ms) and improves insulin sensitivity (Parafati et al., 2018), likely favoring the transport of fat from liver to adipose stores, which does not help weight loss. In fact, we observed only a slight reduction of body mass in this experiment which is not statistically significant. We summarized these observations in the discussion section, lines 426 to 435.

These data confirm our previous data when the final body mass decreased in CAF+BPF group, but the statistical significance was detected only by some tests (T test), but not by one-way ANOVA (Parafati et al., 2015). Similarly, BPF had only weak and not statistically significant effect on body weight in another rodent model of NASH (Musolino et al., 2020). In clinical trials, even if the BPF treatment efficiently reduced TGL and cholesterol levels, the body weight reduction was reported, only if BPF was supplemented with pectins or Cynara cardunculus (Capomolla et al., 2019; Ferro et al., 2020). Our observations are in line with another study, that shows a potent effect of Polyphenol-Rich Rutgers Scarlet Lettuce extract on NAFLD induced with high-fat diet and treated with low-fat diet and no effect on body weight (Cheng et al., 2014). In humans, isocaloric diet modifications also improve NAFLD, but have little effect on body weight (Della Pepa et al., 2017).  

  • The CAF diet produces an evident liver fat accumulation in a grade of micro and macro-steatosis; however, the induction of lipogenesis genes, Fasn, Acly and Acaca, was not statistically significant. How authors explain this?

We explained this phenomenon by short-term fasting, which was applied as a necessary step to measure blood triglycerides and cholesterol levels. This is discussed now in lines 399-414, as follows: However, there was only a weak or not statistically significant increase in transcripts coding for lipogenesis enzymes (Acly, Acaca, Fasn, Scd-1) in CAF livers. This is in contrast to fructose-induced rat and murine models of NAFLD, where the main targets of SREBP-1c- mediated transactivation Acaca and Fasn, were upregulated 6 to 16-fold compared to control animal [23, 51]. Yet, such a strong difference between those models might be explained by overnight fasting, but continued fructose supply before sacrifice [52], while in our study all the animals were deprived for 5 h of all energy sources for blood analysis. Indeed, a study reported that in High-fat diet (HFD)-fed mice Acaca was more expressed in HFD than in the control group and decreased to baseline levels in fasted control animals. In the same experiment, Fasn upregulation by 3 to 4-fold in HFD mice was less dependent on fasting [25]. Accordingly, we found consistent, but modest upregulation of Fasn mRNA in the CAF group by around 1.6, but less consistent data for Acaca and Acly in both array pool (Figure 4) and individual assays (Figure 7), suggesting that lipogenesis enzymes expression is flexible and quickly responds to nutrient status. In contrast, the upstream transcription factors, such as Srebf1 and Pparg maintained the stable 2-fold upregulation, regardless of short-term fasting [25]. 

  • Specifically, what would be the source of malonyl-CoA and free fatty acids if the enzymes activities involved in lipogenesis are not increased?  

According to what we suggested in the answer to the comment 2 above, the levels of these enzymes are regulated by food supply, thus we expect these enzymes would be more induced in CAF group if animals had been not starved for 5 h. Thus, it is plausible that Acly, Acaca and Fasn levels are much higher in fed rats than in fasted ones. 

  • It is established that BPF is hepatoprotective but has no or little effect on obesity. Why specific mechanism allowed by the flavonoids fraction could provide specificity for the damaged liver tissue?

Many natural polyphenol mixtures including BPF have a strong hepatic activity and less effect on adipose tissue. As explained in our answer to p.1, BPF increases insulin sensitivity and this might explain why fat produced in the liver is sent to adipose stores. On the other hand, the bioactive compounds of BPF, which are flavonoids are known to be heavily metabolized by microbiota in the gut, then in enterocytes and finally in the liver, where flavonoids are degraded by phase I metabolism and/or transformed into glucoronates and sulfonated forms. Only very low amounts of active flavonoid metabolites get to the blood stream and may reach the adipose tissue (Spigoni et al., 2017). Thus, the effects of flavonoids are observed more in liver metabolism, but much less in adipocytes.

  • Authors say that BPF induced the Cytochrome 2e1 (CYP2E1), but not by CAF diet feeding without BPF. Then, it is possible that CYP2E1 could play a role in polyphenol metabolism, and its induction is a hepatic response to flavonoids and related metabolites. In fact, what’s known about the pharmacodynamic of BPF? The metabolism of BPF is mainly hepatic? In the case of the Alcoholic Fatty Liver Disease, it is possible to find a catabolic competition between alcohol and BPF?

We thank the reviewer 1 for this comment, since it helped us to

possible explanation for CYP2E1 gene expression induction by BPF in our NAFLD model. First of all, clinical studies have shown that CYP2E1 mRNA and protein levels were decreased with NAFLD progression (Fisher et al., 2009), supporting that CYP2E1 upregulation is not a marker of NAFLD, but rather of Alcoholic Fatty Liver Disease (AFLD) in humans. We do not know what cytochromes are involved in BPF metabolism, but the literature suggests that flavonoids abundant in BPF, such as naringenin, hesperidin and apigenin are subjected to phase I metabolism, involving hydroxylation and demethylation by different cytochromes, including CYP2E1 (Benkovic et al., 2019). In addition, many flavonoids cause reversible inhibition of CYP2E1 as well as some other cytochromes, such as CYP1A2, CYP2A6, CYP2C8 and CYP3A4 (Boniface et al., 2022). Thus, the treatment with BPF may inhibit CYP2E1 activity and this may in turn mediate the induction of CYP2E1 gene expression by a feedback mechanism to partially compensate for the loss of CYP2E1 activity, required for hepatic flavonoid metabolism. However, in case of AFLD, where alcohol induces very high CYP2E1 levels, BPF might counteract these effects. Of course, specific studies should be performed to definitively answer this question, but it is likely since other studies show inhibition of CYP2E1 activity and expression by apigenin in alcohol-induced fatty liver. We mentioned briefly these findings in the discussion in lines 470-474.

Reviewer 2 Report

There are some positive notes in this research. There dietary intervention clearly incudes increased lipid deposition the liver associated with strong induction in lipogenic gene transcription. However, given that no other liver pathologies were reported, I am skeptical to say this model is a MASLD/MASH model. Also, I have deep concerns with the methodology around the use of RT2-PCR, especially with CDNA pooling and the lack of statistical analysis with the data.

1. Line 14 in the Abstract, please remove the comma after flavonoids.

2. Line 153: bouncer should be corrected into douncer.

3. Discussion: Line 404-405, the authors did not measure parameters for glucose homeostasis. Thus, the statement was not 100% supported by data.

Author Response

Rebuttal letter to reviewers’ comments on Ms ID: antioxidants-2976075

We thank the editor and reviewers for their constructive comments that helped us to improve the quality of our manuscript. We addressed all comments and performed new experiments (qRT-PCR analysis for other 7 genes) with the aim to verify our findings according to reviewers’ suggestions. In particular, we explained the rationale behind pooling the RNA samples for RT2-PCR arrays and supported by a vast literature on the topic. In addition, we made changes in the text and in the figures (2, 3, 4, 6, 7 and 8) as requested by the reviewers to further improve the clarity and data presentation. Supplementary Information file was also improved. Finally, we modified slightly the title to respond, at least in part to reviewers’ request.

For clarity we accepted all the changes in the revised final version of our manuscript Ms (MsR1) but provide an additional version where all the changes have been marked automatically by the Microsoft Office Word tracking system.

Please, see the point to point reply to your comments:

We thank the review nr 2 for the detailed review and critical comments, that have helped us to improve our Ms.

  • The introduction is very long which includes a detailed paragraphs on lipogenesis, beta-oxidation, autophagy. I suggest that the authors summarize these paragraphs.

Thank you for this suggestion. Following your recommendation, we reduced the previous introduction from 1052 words down to 750 words, by reducing unnecessary details in all paragraphs, except the last one. 

 I have concerns on the following aspects of the methods.

  • What was the reason why the sample size is considerably small (n=5/group)?

Based on our previous experience, involving 7 to 8 animals per group (Parafati et al., 2015; La Russa et al., 2019), 5 animals would be sufficient to show statistically significant differences for the most measured parameters. Our present data confirm this finding. The only parameter that might require higher rat numbers to detect any statistical difference between CAF and CAF+BPF groups is the final body mass, but as discussed in the answer to R1p1, these differences in body mass are not significant even with 8 animals per group. 

  • Why cDNA samples were pooled together into 1 in the RT2-PCR? My concern with this approach is that statistics can never be done is the sample size is 1. How did the authors come up with 1.7 fold change as the threshold for significant difference?

We thank the reviewer for the opportunity to clarify the reason behind pooling three RNA samples in our microarray approach. As mentioned now in the revised Ms (line 266-268), pooling RNA samples has been validated in many studies as a cost-effective approach to reduce the variability and compensate for the loss of the number of replicates in microarray experiments (Peng X, Wood CL, Blalock EM, Chen KC, Landfield PW, Stromberg AJ. Statistical implications of pooling RNA samples for microarray experiments. BMC Bioinformatics. 2003 Jun 24;4:26. doi: 10.1186/1471-2105-4-26). In addition, recent evidence by Ko, B. et collegues (Ko, B.; Van Raamsdonk, J.M. RNA Sequencing of Pooled Samples Effectively Identifies Differentially Expressed Genes. Biology 2023, 12, 812) highlights the importance of pooling individually isolated RNA samples prior analysis as effective at identifying differentially expressed genes as RNA samples individually.

The technical replicates, if possible, improve the reliability of the data and correct for a possible pipetting error.  We performed three independent RT2 arrays on control group pooled cDNA samples and 2 arrays for each cDNA pool of CAF and CAF+BPF groups. This approach does not allow to test for the biological variability but provides a reliable estimate of differentially expressed genes as explained above.

The threshold for significant difference can be adjusted arbitrarily, but usually it is between 1.5 and 2.0. The software allowed us to choose 1.7 fold change. We selected this threshold as several genes were induced just below 2 fold, but more than 1.5. 

  • For consistency, can the authors also use the ΔΔCT method with normalization for the qRT-PCR.

The reason why the expression was calculated by 2^ΔCT method and SC (control expression) value was not converted to 1 in figure 7, was the following: we believe it provides more information. In fact, it shows what is the relative expression level of each analyzed gene with respect to the house-keeping gene (Hrpt), i.e. it reveals the differences in expression between different genes. To compensate for the lack of normalization to 1, which shows the fold change with respect to control group, we indicated fold change values by colored numbers directly on the graphs in figure 7, showing them with respect to SC and CAF mean indicated.  

  • Also for consistency, use scatter plots instead of bar graphs.

As requested we converted all graphs in scatter graphs except for the graphs showing Western blotting data in Fig. 8. 

  • What is/are the reason/s why the authors did not include all rats in the Wb analysis?

All rats were included in the WB analysis, but we have shown only one representative gel with 3 animals, because we could not load all 5 lysates per group on the same gel. We did not have 16-slot gels. The second WB gel was shown in the Supplementary data file.

  • Can the authors characterize the type of lipids which accumulated in the liver. In the blood, on TG, but not Cholesterol, was increased. Would this be the same in the liver?

This is an interesting suggestion, but type of lipids contained in lipid droplets of CAF vs CAF+BPF livers can be nicely characterized by metabolomic analysis, but it is out of scope of our Ms.

In this animal model (Wistar rats) we could not detect higher cholesterol blood levels in CAF group with respect to controls after 14 weeks of treatment, which is consistent with our previous data (Parafati et al., 2015).

  • Can the authors evaluate for liver pathologies as as inflammation, balooning to see if the rat model was a MASLD model?

As requested we evaluated the inflammation by performing RT-qPCR  on the following cytokines: Il6, Il10, Tnf, Ilb1 and Infg. The results suggest a significant induction of Il6, Il10, Il1b and TNF and significant downregulation of these cytokines by BPF (except Il1b). These data were included in the new figure 7.

The infiltration of Kupffer and other inflammatory cells was demonstrated in our previous study in the same model at the same time of CAF diet application (Parafati et al., 2015). The ballooning is rare after 3,5 months but was evident in rats exposed to CAF diet for 7 months (Parafati et al., 2018). Our data suggest that rats develop an initial stage of NASH after 14 weeks of CAF diet application.

  • What is the reason why TUBA and GAPDH were interchangeably used as loading protein in WB?

The blots in fig. 8 A and B were performed at 2 different times and we used the best loading control antibody that was available at that time. 

  • For Figure 9, I suggest the authors integrate BPF in the figure. Also remove details that were not measured. Concentrated on triglycerides since it the lipid that is changed by BPF.

Thanks for this suggestion. We have integrated BPF in figure 9 and removed unnecessary details, but tried to include all modulated genes.  

Detail comments:

  1. Line 14 in the Abstract, please remove the comma after flavonoids.

Done. Thanks for pointing out this mistake.

  1. Line 153: bouncer should be corrected into douncer.

Done. Thanks for pointing out this mistake.

  1. Discussion: Line 404-405, the authors did not measure parameters for glucose homeostasis. Thus, the statement was not 100% supported by data.

Thanks for pointing out this mistake. In fact, in this study we did not measure glycemia. This was done in previous studies with rats (La Russa et al., 2019; Parafati et al., 2015). The word “diabetes” was unnecessary in this statement, and it was deleted.

Reviewer 3 Report

The application of mRNA PCR array profiling to study metabolism and autophagy-related gene expression, alongside investigating the effects of a Bergamot Polyphenol Fraction (BPF) diet on Non-Alcoholic Fatty Liver Disease (NAFLD), is a promising area of research that could benefit future treatments. The author aimed to understand the molecular mechanisms that underlie the development and potential reversal of cafeteria diet-induced hepatic steatosis. To confirm key findings, additional protein measurements were conducted on some genes that were differentially expressed. Below are a few recommendations for your consideration:

Main comments

• Why was a pooled RNA strategy chosen for the experimental design?

• Please discuss the limitations of the PCR array and small sample size.

Please tune down the claim in the title due to only limited gene expression was performed in the study.

On pg. 4, line 159, only 500 ng of pooled RNA was used for RT followed by mRNA quantification, and the experiment was repeated. Why was such a low amount of RNA used for quantification?

• Line 182, the description is inaccurate. Relative mRNA expression was calculated using 2^(-∆∆CT), which is a power calculation; the fold change is a comparison of the relative expression between any two samples.

• Fig. 4 and Fig5 need to be modified as all text is not legible. Why was a log10 transformation used?

• What additional information do we gain from Fig. 6? It compares the fold change between CAF/SC and CAF+BPF/SC. Thus, the results should be the same as comparing CAF vs CAF+BPF.

• The titles of Fig. 5 and Fig. 4A are inconsistent.

• The choice of the word “regulation,” e.g., line 481, is questionable. The author did not perform any experiments on regulation; the current data only showed differentially expressed genes between two groups with a positive or negative fold change. Similarly, in Fig. 4 and 5, the term “modulated” suggests a direct interaction between genes, which is not established.

Author Response

Rebuttal letter to reviewers’ comments on Ms ID: antioxidants-2976075

We thank the editor and reviewers for their constructive comments that helped us to improve the quality of our manuscript. We addressed all comments and performed new experiments (qRT-PCR analysis for other 7 genes) with the aim to verify our findings according to reviewers’ suggestions. In particular, we explained the rationale behind pooling the RNA samples for RT2-PCR arrays and supported by a vast literature on the topic. In addition, we made changes in the text and in the figures (2, 3, 4, 6, 7 and 8) as requested by the reviewers to further improve the clarity and data presentation. Supplementary Information file was also improved. Finally, we modified slightly the title to respond, at least in part to reviewers’ request.

For clarity we accepted all the changes in the revised final version of our manuscript Ms (MsR1) but provide an additional version where all the changes have been marked automatically by the Microsoft Office Word tracking system.

Please see details below:

Main comments

  • Why was a pooled RNA strategy chosen for the experimental design?

Thanks for this comment as it allows us to address the importance of RNA pooling in microarray experiments. Vast experimental evidence suggests that RNA sample pools are the best way to reduce the biological variability and compensate for the loss of the number of replicates (Takele Assefa et al., 2000 https://bmcgenomics.biomedcentral.com/articles/10.1186/s12864-020-6721-y). This has been shown to be equally true for microarray experiments and sample pooling has become a routine approach in many laboratories (Peng X, Wood CL, Blalock EM, Chen KC, Landfield PW, Stromberg AJ. Statistical implications of pooling RNA samples for microarray experiments. BMC Bioinformatics. 2003 Jun 24;4:26. doi: 10.1186/1471-2105-4-26). Thus, RNA pooling provides equivalent power and improves efficiency and cost-effectiveness of differential expression analysis. The technical replicates, if possible, improve the reliability of the data and correct for a possible pipetting error.  We performed three independent RT2 arrays on control cDNA samples from pooled totRNA of SC group and 2 arrays for each cDNA pool of CAF and CAF+BPF groups.

  • Please discuss the limitations of the PCR array and small sample size.

The PCR array is limited to 84 genes and does not allow to validate the changes in the expression of other genes that might be important in the analyzed process. It is also more prone to technical errors and possible differences in the efficiency of cDNA synthesis and PCR amplifying primers than RNA-seq experiments. For this reason, it is more difficult to validate small expression changes that might be relevant in a tested biological phenomenon. In our approach, we combined the technical replicates of the microarray approach with pooling RNA samples of three more representative rats (for weight and liver histology) to reduce the impact of a technical error. Of course, the pooling approach does not allow to test for biological variability, which is quite significant. To test for biological variability, we performed independent qRT-PCR assays on RNA extracted from individual animals. Since the number of individual RNA samples in some cases was 4, the variability was significant and statistical differences could be detected only with less stringent tests.   

  • Please tune down the claim in the title due to only limited gene expression was performed in the study.

We are confident that sufficient evidence has been provided for a powerful lipogenesis suppression in BPF-treated animals (RT2-PCR array, qRT-PCR and Western blot data) and at least for two b-oxidation genes Cpt1b and Ppara at the level of gene expression, confirmed by an independent qRT-PCR. No other B-oxidation gene, analyzed on the array does seem to be induced.  However, following the reviewer 3 suggestion we added the word “among” to tune down the claim that these pathways are the only main effects of BPF treatment, as other biological processes are also affected.  

Detail comments:

  • On page 4, line 159, only 500 ng of pooled RNA was used for RT followed by mRNA quantification, and the experiment was repeated. Why was such a low amount of RNA used for quantification?

We verified the description in material and methods and made clearer statements to avoid misinterpretation: RNA quantification was carefully performed before pooling and 500 ng of pooled RNA was used for each microarray plate according to manufacturer instructions, but it was scaled up for 2 or 3 plates. See corrections in lines 131 to 135.  

  • Line 182, the description is inaccurate. Relative mRNA expression was calculated using 2^(-∆∆CT), which is a power calculation; the fold change is a comparison of the relative expression between any two samples.

According to the definitions of the Gene Globe Data Analysis software “Fold-Change (2^(- Delta Delta CT)) is the normalized gene expression (2^(- Delta CT)) in the Test Sample divided the normalized gene expression (2^(- Delta CT)) in the Control Sample. Therefore, 2^(- Delta Delta CT) = (2^(- Delta CT)) in the Test Sample/(2^(- Delta CT)) in the Control Sample. We actually modified the last part of the § Gene expression analysis regarding RT2-PCR data analysis in the following way: (now from line 148 to 157 ) “This and further analysis was performed by using a dedicated software available at the Gene Globe Data Analysis Center (http://www.qiagen.com/it/shop/genes-and-pathways/data-analysis-center-overview-page/). Briefly, the raw cycle threshold (CT) data of replicate plates for each experimental group (SC, CAF or CAF+BPF) were subjected to quality control, according to internal plate controls and standard parameters. Subsequently, two hk control genes—hypoxanthine phosphoribosyl transferase (Hprt1) and B2-microglobulin (B2m) were selected for normalization ….”

Fig. 4 and Fig 5 need to be modified as all text is not legible. Why was a log10 transformation used?

We prepared better resolution figures and expanded them to make the data easily legible.

The log10 transformation is the form of data presentation generated routinely by Gene Globe Data Analysis Center software used for RT2 array data analysis (see material and methods).

  • What additional information do we gain from Fig. 6? It compares the fold change between CAF/SC and CAF+BPF/SC. Thus, the results should be the same as comparing CAF vs CAF+BPF.

Thanks for this remark. In fact, the data CAF vs SC should be the same as the data presented in Fig. 4A to C. However, the figure 4D-F does not compare CAF+BPF group with SC group, but only CAF+BPF with CAF. Thus, showing CAF and CAF+BPF vs SC on the same graph allows us to appreciate the effect of BPF with respect to CAF and SC at the same time.

In addition, the column graphs in Fig. 6, which was modified for this revision allowed us to implement additional information regarding the functional division of genes and the standard deviation in triplicate control samples and the replicate array plates of test groups, which were calculated based on the standard deviation of DeltaCT. SD has been added as the error bars to the new version of Figure 6, now on the page 10.

  • The titles of Fig. 5 and Fig. 4A are inconsistent.

As requested we corrected both figure legends to make them consistent and we also substituted the term “gene expression regulation” with “differentially expressed genes”.

  • The choice of the word “regulation,” e.g., line 481, is questionable. The author did not perform any experiments on regulation; the current data only showed differentially expressed genes between two groups with a positive or negative fold change. Similarly, in Fig. 4 and 5, the term “modulated” suggests a direct interaction between genes, which is not established.

Thank you for pointing out this incongruency. The excessive use of the term “gene regulation” was revised in the text and in the legends of figure 5 and 4A, which were corrected as suggested.  However, we would like to draw attention to the fact that the term “regulation” is used in its figurative meaning. In fact, the term “fold regulation” has been formulated   to indicate any type of difference in gene expression and is routinely used by Gene Globe Data Analysis beside other software.

Benkovic, G., M. Bojic, Z. Males, and S. Tomic. 2019. Screening of flavonoid aglycons' metabolism mediated by the human liver cytochromes P450. Acta Pharm. 69:541-562.

Boniface, P.K., F.B. Fabrice, H.K. Paumo, and L.M. Katata-Seru. 2022. Protective roles and mechanism of action of plant flavonoids against hepatic impairment: Recent developments. Curr Drug Targets.

Capomolla, A.S., E. Janda, S. Paone, M. Parafati, T. Sawicki, R. Mollace, S. Ragusa, and V. Mollace. 2019. Atherogenic Index Reduction and Weight Loss in Metabolic Syndrome Patients Treated with A Novel Pectin-Enriched Formulation of Bergamot Polyphenols. Nutrients. 11.

Cheng, D.M., N. Pogrebnyak, P. Kuhn, A. Poulev, C. Waterman, P. Rojas-Silva, W.D. Johnson, and I. Raskin. 2014. Polyphenol-rich Rutgers Scarlet Lettuce improves glucose metabolism and liver lipid accumulation in diet-induced obese C57BL/6 mice. Nutrition. 30:S52-58.

Della Pepa, G., C. Vetrani, G. Lombardi, L. Bozzetto, G. Annuzzi, and A.A. Rivellese. 2017. Isocaloric Dietary Changes and Non-Alcoholic Fatty Liver Disease in High Cardiometabolic Risk Individuals. Nutrients. 9.

Ferro, Y., T. Montalcini, E. Mazza, D. Foti, E. Angotti, M. Gliozzi, S. Nucera, S. Paone, E. Bombardelli, I. Aversa, V. Musolino, V. Mollace, and A. Pujia. 2020. Randomized Clinical Trial: Bergamot Citrus and Wild Cardoon Reduce Liver Steatosis and Body Weight in Non-diabetic Individuals Aged Over 50 Years. Front Endocrinol (Lausanne). 11:494.

Fisher, C.D., A.J. Lickteig, L.M. Augustine, J. Ranger-Moore, J.P. Jackson, S.S. Ferguson, and N.J. Cherrington. 2009. Hepatic cytochrome P450 enzyme alterations in humans with progressive stages of nonalcoholic fatty liver disease. Drug Metab Dispos. 37:2087-2094.

La Russa, D., F. Giordano, A. Marrone, M. Parafati, E. Janda, and D. Pellegrino. 2019. Oxidative Imbalance and Kidney Damage in Cafeteria Diet-Induced Rat Model of Metabolic Syndrome: Effect of Bergamot Polyphenolic Fraction. Antioxidants (Basel). 8.

Musolino, V., M. Gliozzi, F. Scarano, F. Bosco, M. Scicchitano, S. Nucera, C. Carresi, S. Ruga, M.C. Zito, J. Maiuolo, R. Macri, N. Amodio, G. Juli, P. Tassone, R. Mollace, R. Caffrey, J. Marioneaux, R. Walker, J. Ehrlich, E. Palma, C. Muscoli, P. Bedossa, D. Salvemini, V. Mollace, and A.J. Sanyal. 2020. Bergamot Polyphenols Improve Dyslipidemia and Pathophysiological Features in a Mouse Model of Non-Alcoholic Fatty Liver Disease. Sci Rep. 10:2565.

Parafati, M., A. Lascala, D. La Russa, C. Mignogna, F. Trimboli, V.M. Morittu, C. Riillo, R. Macirella, V. Mollace, E. Brunelli, and E. Janda. 2018. Bergamot Polyphenols Boost Therapeutic Effects of the Diet on Non-Alcoholic Steatohepatitis (NASH) Induced by "Junk Food": Evidence for Anti-Inflammatory Activity. Nutrients. 10.

Parafati, M., A. Lascala, V.M. Morittu, F. Trimboli, A. Rizzuto, E. Brunelli, F. Coscarelli, N. Costa, D. Britti, J. Ehrlich, C. Isidoro, V. Mollace, and E. Janda. 2015. Bergamot polyphenol fraction prevents nonalcoholic fatty liver disease via stimulation of lipophagy in cafeteria diet-induced rat model of metabolic syndrome. J Nutr Biochem. 26:938-948.

Spigoni, V., P. Mena, F. Fantuzzi, M. Tassotti, F. Brighenti, R.C. Bonadonna, D. Del Rio, and A. Dei Cas. 2017. Bioavailability of Bergamot (Citrus bergamia) Flavanones and Biological Activity of Their Circulating Metabolites in Human Pro-Angiogenic Cells. Nutrients. 9.

Wu, S., J. Tan, H. Zhang, D.X. Hou, and J. He. 2023. Tissue-specific mechanisms of fat metabolism that focus on insulin actions. J Adv Res. 53:187-198.

Round 2

Reviewer 2 Report

See detailed comments

1. Although the authors have cited two articles where pooling of RNA samples were utilized for microarray or RNA sequencing, these manuscripts still have replicates so that proper statistical analysis can be performed. In contrast, in this manuscript, there is one biological sample, preventing statistical analysis. 

2. I disagree with the authors that using 2 ways of expressing gene expression changes provides more information. In fact, it is confusing. The purpose of the subsequent RT-PCR is to confirm the microarray. Thus, expressing them in the same "fold-change" manner will make the data more coherent.

3. The authors mention that there were n=5/group. How come there are 6 replicates in Figure 2F?

4. I again disagree with the authors that cholesterol can only be measured using lipidomics/metabolimics. There are commercial plate based assays to measure cholesterol.

Author Response

Rebuttal letter R2 to reviewers’ comments to the first revision of Ms ID: antioxidants-2976075

We thank and reviewer 2 and the editors for their constructive comments that helped us to improve further the quality of our manuscript. We addressed all reviewer’s comments point by point.

  1. Although the authors have cited two articles where pooling of RNA samples were utilized for microarray or RNA sequencing, these manuscripts still have replicates so that proper statistical analysis can be performed. In contrast, in this manuscript, there is one biological sample, preventing statistical analysis. 

R1: We are sorry, but we were not clear enough in explaining that we performed 3 technical replicates for the control group and two technical replicates for CAF and two replicates for CAF+BPF group. This was stated in Material and Methods section, but also in our response to the point #3 of reviewer’s comments, where it was probably not sufficiently comprehensible: The technical replicates, if possible, improve the reliability of the data and correct for a possible pipetting error.  We performed three independent RT2 arrays on control group pooled cDNA samples and two arrays for each cDNA pool of CAF and CAF+BPF groups. This approach does not allow to test for the biological variability but provides a reasonable estimate of differentially expressed genes.  

Of course, duplicates are not sufficient for reliable statistics as in case of treated groups, but they are mathematically enough to calculate the standard deviation (see comments in Graphpad) https://www.graphpad.com/support/faqid/591/#:~:text=Is%20valid%20to%20calculate%20the,(N%3D2)%20data). In fact, we performed the calculation of SDs for all experimental samples and included them in the graphs in the Figure 6 in the revised R1 version. The graph shows that in some cases SDs are very small and they do not overlap between groups, while in other cases SDs are very large and they overlap, suggesting that these data should be interpreted with caution. For this reason, it was necessary to confirm RT2-PCR array data by independent Rt-qPCR analysis, which was done for 16 candidates differentially expressed genes as stated now in lines 323-25: This approach was useful to assess biological variability, lost by pooling RNAs and to compensate for limited number of technical replicas in RT2-PCR array analysis.

In addition, to address reviewer’s 2 criticism we stated in the discussion that some RT2-PCR array data should be interpreted with caution due to limited number of technical replicates (lines 391-393).

  1. I disagree with the authors that using 2 ways of expressing gene expression changes provides more information. In fact, it is confusing. The purpose of the subsequent RT-PCR is to confirm the microarray. Thus, expressing them in the same "fold-change" manner will make the data more coherent.

R2: Yes, it is true, the purpose of RT-qPCR is to confirm the RT2 array data expressed in fold change. Thus, for the sake of clarity, as suggested by Reviewer 2, we re-calculated the data as fold-change and substituted the relative expression data set (now in Fig. S1, Supplementary material) with the new set of fold-change graphs as the main figure 7.

  1. The authors mention that there were n=5/group. How come there are 6 replicates in Figure 2F?

R3: Thank you for pointing out this inconsistency. We had 5 animals and at least 1 lysate for each animal, but 6 slots to load, therefore we loaded an additional lysate for each group, which was included in the statistics. We agree that it may cause a bias in the data and we eliminated this point.

  1. I again disagree with the authors that cholesterol can only be measured using lipidomics/metabolimics. There are commercial plate based assays to measure cholesterol.

R4: Reviewer 2 is absolutely right. Cholesterol content in the hepatic tissue can be measured by many methods. However, since we are planning to assess lipid/sterols profile by lipidomics in this animal model this interesting issue could be properly addressed and discussed in a separate work. Thus, we believe that solving this issue here in the context of differential gene expression induced by CAF diet and BPF is out of scope of this paper.
